# Proteome-Based Serotyping of the Food-Borne Pathogens *Salmonella Enterica* by Label-Free Mass Spectrometry

**DOI:** 10.3390/molecules27144334

**Published:** 2022-07-06

**Authors:** Xixi Wang, Chen Chen, Yang Yang, Lian Wang, Ming Li, Peng Zhang, Shi Deng, Shufang Liang

**Affiliations:** 1State Key Laboratory of Biotherapy and Cancer Center, West China Hospital, Sichuan University, and National Collaborative Innovation Center for Biotherapy, Chengdu 610041, China; pizzawx@sina.cn; 2Chengdu Center for Disease Control and Prevention, Chengdu 610041, China; yanger326@163.com (Y.Y.); septwolvesnjwl@163.com (L.W.); lm781227@163.com (M.L.); 3Regenerative Medicine Research Center, West China Hospital, Sichuan University, Chengdu 610041, China; cc10027@163.com; 4Department of Urinary Surgery, West China Hospital, West China Medical School, Sichuan University, Chengdu 610041, China; zpeng2001@163.com (P.Z.); dengshidiy@163.com (S.D.)

**Keywords:** proteome, peptide markers, *Salmonella enterica* serovars, strain similarity, C5.0 decision tree

## Abstract

Food-borne diseases caused by *Salmonella enterica* of 2500 serovars represent a serious public health problem worldwide. A quick identification for the pathogen serovars is critical for controlling food pollution and disease spreading. Here, we applied a mass spectrum-based proteomic profiling for identifying five epidemiologically important *Salmonella enterica subsp. enterica* serovars (*Enteritidis*, *Typhimurium*, *London*, *Rissen* and *Derby*) in China. By label-free analysis, the 53 most variable serovar-related peptides, which were almost all enzymes related to nucleoside phosphate and energy metabolism, were screened as potential peptide biomarkers, and based on which a C5.0 predicted model for *Salmonella enterica* serotyping with four predictor peptides was generated with the accuracy of 94.12%. In comparison to the classic gene patterns by PFGE analysis, the high-throughput proteomic fingerprints were also effective to determine the genotypic similarity among *Salmonella enteric* isolates according to each strain of proteome profiling, which is indicative of the potential breakout of food contamination. Generally, the proteomic dissection on *Salmonella enteric* serovars provides a novel insight and real-time monitoring of food-borne pathogens.

## 1. Introduction

*Salmonella*, more than 2500 serotypes, is a major zoonotic food-borne pathogen, which causes outbreaks and sporadic cases of gastroenteritis in humans. Approximately 300 serovars are reported in China, of which *Enteritidis*, *Typhimurium*, *Rissen*, *Derby*, and *London* are among the top five *Salmonella enterica* serovars isolated from food-borne Salmonella infections. The *Salmonella* strains have a drastic change in virulence or expression under the condition of a single gene mutation. Therefore, the identification and characterization of species and subspecies are generally necessary for pathogen confirmation and clinical diagnostics. Moreover, food safety efforts require serovar and strain-level specificity in order to trace back the source of bacterial contamination.

The *Salmonella* serotyping method based on the White–Kauffmann–Le Minor scheme is accepted worldwide as a gold standard for the differentiation of *salmonella* below the subspecies level [1]. It is determined by a combination of biochemical reactions and serotyping of the somatic O, flagellar H, and capsular Vi antigens. However, the antigen-based serotyping, often performed by slide agglutination, is laborious, time-intensive, and expensive, due to more than 200 different antisera and the wide-ranging quality of antibodies.

The pulsed field gel electrophoresis (PFGE) profiling has become a gold standard for the molecular subtyping of *Salmonella*, and PCR-based amplification and genetic sequencing are also becoming increasingly popular for strain identification. However, both of the two approaches have difficulty in precisely distinguishing two highly similar serovars such as *S. Typhimurium* and *S. Heidelberg*. Other DNA-based techniques, including plasmid profile, ribosomal DNA intergenic spacer amplification, multi-locus sequence typing (MLST), multi-locus variable numbers tandem repeat analysis (MLVA), and clustered regularly interspaced short palindromic repeats (CRISPR), are usually available, which are still challenging due to the inefficiency of indicating phenotypes and multiple primers required for amplifying the sequences of untargeted genes.

Mass spectrometry (MS) provides a high-throughput and relatively unbiased view of the protein profiling in bacteria [2], which facilitates the differentiation of genetically-related bacteria and decoding the new nonsynonymous single-nucleotide polymorphisms. In recent years, a technique known as direct bacterial profiling has increasingly been applied to dissect proteome for bacterial species identification via the matrix-assisted laser desorption ionization–time of flight mass spectrometry (MALDI-TOF MS) [3,4,5,6,7,8,9,10]. A sufficient number of stable mass signals of major housekeeping proteins, such as ribosomal proteins, are reproducibly detected for bacterial species identification by using simple mass pattern-matching approaches or more sophisticated algorithms to compare and estimate the similarities between spectra [11,12,13,14,15,16,17,18]. These characteristic mass profiles (patterns) were successfully applied to bacterial subtyping at the species level, but hardly recognized the serovar level (i.e., the H and O antigen levels) [19] due to their bias toward small ribosomal proteins with a limited range of 2−20 kDa.

With the wider mass range, better sample to sample reproducibility, and greater number of proteins ionized, the electrospray ionization (ESI)-based MS platform provides access to a more diverse range of proteins, potentially providing greater specificity for bacteria [20,21,22]. It has already been used to identify markers that differentiate thermophilic versus nonthermophilic groups of *Cronobacter sakazakii* [23], identify proteins characteristic of specific outbreak strains of *V. parahaemolyticus*, guide the development of PCR probes [19], and to differentiate closely-related species within the *enterobacteriaceae* family [22,24]. In addition, the approach has been shown to quantify protein expression differences by using certain housekeeping proteins as internal standards.

Data mining science can extract useful knowledge which is hidden in the data. Predictive analytics is the process by which information is extracted from existing data sets for determining patterns and predicting the forthcoming trends or outcomes. For clinical trials, using the extracted prediction rules has the potential to predict the pathogenic factors [25], disease progression, and prognosis [26]. Among the different methods of data mining, the decision tree is one of the most powerful and common tools for creating predictions.

In this article, we demonstrate a proteomic analysis for identifying serovar-specific peptide markers among *Enteritidis*, *Typhimurium*, *London*, *Rissen*, and *Derby*. Fifty-three of the most variable serovar-related peptides have been identified as potential biomarkers. Taking the 53 peptides as variables and serotypes as target, a C5.0 predicted model with four predictor peptides was generated. A test set of 15 *Salmonella enterica* strains were then classified with the accuracy of 94.12%. It has also turned out that is it effective to apply the whole proteomic profiling process to determine the genotypic similarity among *Salmonella enteric* isolates as a sign of the breakout of a food contamination incident by comparing the results to PFGE.

## 2. Results

### 2.1. Comprehensive Proteomic Profiling for Different Salmonella Serotypes

In the UniprotKB database for *Salmonella*, there was redundancy and uncertainty on proteins and their peptide sequences. Instead of the proteins, we applied peptides to achieve more veracity.

Forty *Salmonella* isolates with different serotypes were divided into the training and testing groups (Table 1). The training group was analyzed by LC-MS/MS-based proteomics. The total 7339 peptides were identified (FDR > 0.01) and quantified by label-free quantitation (LFQ) in at least two technical repeats. For repeated peptides, due to the digested or mechanical cleavages, the longer sequenced peptides were preferred. To avoid the peptide markers’ variance, we further limited the acceptable peptides as being identified in all replicates within at least one certain serotype. In addition, 1027 peptides from 660 proteins were used for marker development, of which 108 peptides were found in *Enteritidis*, 67 in *Typhimurium*, 239 in *Rissen*, 462 in *Derby*, and 447 in *London*. 

### 2.2. Peptide Markers for Salmonella enterica Serotyping

To discover the peptide markers for the identification of *Enteritidis*, *Typhimurium*, *London*, *Rissen*, and *Derby* in the training set, the Perseus computational platform [27] was applied to filter the 1027 potential peptide markers by reversed identification and potential contamination. Then, the LFQ values of peptides were logarithmized and the imputation was done by normal distribution with a 0.3 width and 1.8 down-shift. A multiple-sample ANOVA test was performed with a permutation-based testing correction that was controlled by using an FDR threshold of 0.05. The coming results showed that 76 peptides that were collected with ANOVA were significant (*p* < 0.05). It was considered that the identified peptides may be acquired by MS randomly in data-dependent acquisition (DDA) and could not be characterized by typical chromatographic peaks to quantify accurately. Therefore, we checked all the significantly changed peptides in Skyline [28] (Figure 1), and only 53 peptides (Appendix A) were certificated with a good peak shape (≥6 acquired data points, and S/N ≥ 10). The power of 53 peptides to separate the five serotypes from each other was shown by a Hierarchical cluster analysis in the training group (Figure 2).

### 2.3. Accuracy of Models and Important Predictor Variables

It still seemed too much for the daily testing work with 53 peptides as a profile pattern for serotyping, so we applied the SPSS Modeler to build a model to predict *Salmonella*
*enterica* serotype. Taking the 53 peptides as the variables and serotypes as the targets, 25 out of 40 *Salmonella* isolates in the training group were processed, and two models, driven by a Quick, Unbiased, Efficient Statistical Tree (QUEST) [29] and C5.0 [30] algorithms with 100% accuracy, were generated for further assessment. The most important predictors based on the C5.0 method were the four peptides, including “IFYNDFQADDADLSDYTNK, YGVVEFDQK, VETISYVK, VANNDLLTILQALK” (Figure 3A). For QUEST, there were also four predictors, which were “AEASQYDALANAR, VETISYVK, YGVVEFDQK, LQYVDESLSDDQVVICGQR” (Figure 3B). The two peptides “VETISYVK, YGVVEFDQK” were common for both methods.

To evaluate the capability of these two models, a testing group, including 15 *Salmonella* isolates, were predicted. The accuracy of the C5.0 and QUEST methods were 94.12% and 88.24%, respectively. For C5.0, 14 of 15 strains were identified correctly, while only one isolate could not be categorized among the five serotypes. The exceptional undetectable one was *Sagona*, which was recognized by the White–Kauffmann–Le Minor scheme. The QUEST model did not recognized *Sagona* and mismatched the *Rissen* to *Eneritidis*. So, the C5.0-based predicted model was more reliable for *Salmonella enterica* serotyping in this research.

### 2.4. Hierarchical Clustering to Differentiate Similarity among Salmonella enteric Isolates

A hierarchical clustering (HCL) of all quantitative peptides was performed in Perseus using Euclidean distances. The distance threshold was defined by the variances of the repetitions of the Quality Control (QC) sample and used for the gene-closed cluster identification to distinguish from variance and difference. The QC sample was produced by randomly pooling the protein extraction from all the isolates and acquiring every 5–10 unknown samples. Twenty-five isolates from the training group were clustered and the results showed that none of the 25 isolates had descended from a common ancestor due to their Euclidean distances comparing to the QC sample (Figure 4). To further test the HCL efficiency for genotypic similarity analysis, six *Enteritidis* isolates extracted from a food poisoning incident were analyzed. All six isolates were determined to have the same ancestor (Figure 5A), which was consistent with PFGE results, except for the strain No. 1210 which had a different gene-type from the other five strains (Figure 5B).

### 2.5. Exploring the Genetic and Biological Explanations for the Distinct Proteomic Profiles among the Salmonella serotypes

When associating the biological process with distinct proteomic profiles in these five *Salmonella* serotypes, we aligned the peptide markers to certain proteins (Appendix A) by following rules in case of redundant proteins: (1) the protein is active by query, and (2) all the redundant protein IDs for one peptide attribute to one single gene. A total of 50 proteins of 53 peptides were finally listed (Appendix A), of which 25 proteins tended to be located at the cellular anatomical entity, and more than 80% of proteins were significantly enriched for the GO (Gene Ontology) molecular function terms of the catalytic and binding activity. These proteins were involved in cellular processes and metabolic processes. 

Among the changed proteins, some candidates belong to enzymes with catalytic and binding activities, which were analyzed against the ExplorEnz database (https://www.enzyme-database.org (accessed on 17 March 2022)). As a result, 27 proteins were aligned to the certain EC (Enzyme Commission) number with an exact name and Reaction equation (the revised Appendix A). The enriched enzyme types include Oxidoreductases, Transferases, and Hydrolases, which confer their functions in the biological process of amino-acid biosynthesis, carbohydrate metabolism, and pyrimidine metabolism. 

The previous studies have revealed that several molecules, including ethanolamine, gluconic acid, monosaccharide, phenylalanine, pipecolic acid, and saccharide, are able to discriminate specific *S. Typhi* species and *S. Paratyphi A* cases [31]. For instance, mercaptoacetic acid and α-hydroxypentanedioic acid are biomarkers of the *S. enteritidis* serotype, while N-acetyl aspartic acid, 2(1H)-pyrimidone, and L-threonic acid are only produced by the *S. typhi* serotype. The tetradecanoic acid methyl ester and glyceryl ether-glucoside are released only by the *S. typhimurium* serotype. 1,4-naphthoquinone is only generated by the *S. choleraesuis* serotype, and the 3-pyridinecarboxylic acid trimethylsilyl ester is unique to the *S. arizonae* serotype [32]. In general, the genetic analysis suggests that the genetic and metabolic determinants of *Salmonella* adaptation to animal sources may have been driven by the natural feeding environment of the animal, distinct livestock diets modified by human and environmental stimuli, and the physiological properties of the animal itself. The functionally annotated mutations associated with animal sources include aspartate ammonia-lyase, 2,3,4,5-tetrahydropyridine-2,6-dicarboxylate N-acetyltransferase, dipeptidase E, and phosphoethanolamine transferase EptC. The metabolic pathways mainly impacted by core genome variants contain C4-dicarboxylate transport, aspartate ammonia-lyase activity, and tetrahydrodipicolinate N-acetyltransferase activity [33]. All of these genetic and metabolic results support that the variability of enzymes is a universal phenomenon among different *Salmonella Enterica* serotypes that are adaptive responses to the external environment.

To further verify the enrichment analysis of the proteomics data, we applied the genetic strategy for the annotation of the other 25 peptides, without the EC index, through the location of conserved domain footprints using the NCBI’s Conserved Domain Database (CDD) [34]. The functional sites could be inferred from these footprints which indicates a local or partial similarity to other characterized proteins. Moreover, 11 out of 25 peptides were queried in different gene super families, including the Lyase_I-like super family, PRK super family, Glyco_tranf_GTA_type super family, and DUF1439 super family (Appendix A), which were concentrated upon enzymes related to energy metabolism.

### 2.6. The Serotype-Specific Proteome and Biological Analysis

For the extra serotypes’ proteomic analysis, we also took a view of intra-serotype difference to evaluated the possible pathway for evolution. Four serotypes, including *Enteritidis*, *Typhimurium*, *Derby*, and *Rissen*, were discussed. All four serotypes’ proteome were significantly enriched for the GO molecular function terms of catalytic and binding activity. The EC number was signed by energy metabolism-related phosphorylation and dehydrogenase activity for *Enteritidis*, nucleoside phosphate transferring for *Typhimurium*, Glucose-related energy metabolism for *Derby*, and nucleoside phosphate metabolism for *Rissen*.

### 2.7. The Specificity for the Enteritidis Challenges

The differentiation of *Enteritidis* challenged the usefulness of PFGE in *Salmonella* subtyping activities. Approximately 45% of serovar *Enteritidis* isolates in PulseNet, which were not epidemiologically related, displayed the same PFGE XbaI pattern [35]. In Figure 4, the *Enteritidis* No. 23 and 2056 were clustered together through PFGE, but the proteomic profiles concluded that these two isolates were not epidemiologically related. The result verified the sensitivity and specificity of our proteome-based serotyping approach compared with PFGE.

## 3. Discussion

Compared with the traditional molecular typing method, such as antigen-based serotyping, LC-MS/MS-based proteomic serotyping could distinguish the new serotypes without any antigen specificity or the coding gene of an antigen being mutated.

The serotyping of *Salmonella* is always accompanied with strain similarity and alignment analysis for the potential breakout of food contamination. Both PFGE and LC-MS/MS profile the certain “fingerprint” of the genome or proteins to distinguish specific serotypes. LC-MS/MS usually takes only up to 3 days to synchronously perform the serotyping and similarity analysis, thus it is more rapid and convenient when compared with the antigen-based PFGE serotyping within 5~7 days. PFGE technology is usually difficult to reliably visualize smaller DNA fragments (e.g., <20.5 kb) [36] and has difficulty in differentiating bands differing by <5–10% in size due to the limited resolution of electrophoresis [37,38]. Additionally, the systematic band shifts usually happen due to gel imperfections or imperfect reproducibility of electrophoretic conditions [39]. In addition, PFGE manipulation is not automated and requires high-level technical expertise, and thus, it is hampered by a low throughput, low robustness, and poor comparability of results between laboratories [40,41,42]. The artifacts of the imagine analysis software may also lead to the misidentification of bands. For LC-MS/MS, we applied FASP and StagTips to process samples which were more stable and standardized. So far, in the shotgun-type proteomic approach, by applying lots of mass signals to target the related peptides or proteins, more sensitivity and resolution on bacteria serotyping are achieved in discriminatory than size-based PFGE. Two sets of *C. perfringens* isolates that have a 100% similarity of PFGE profiles were successfully distinguished using LC-MS/MS identification [43].

From the one thousand and twenty-seven peptides in the LC-MS/MS identification, fifty-four peptides showed two homologous peptides with only one different amino acid, and the other three peptides had three homologous peptides by one different amino acid (Appendix A). All these peptides were changed and specific between at least two *Salmonella* serotypes. It is greatly supported that the difference in the *Salmonella* strains could probably be generated by the replacement of several amino acids. Furthermore, a PFGE-based prediction of these serovars is unreliable if isolates in the database are not representative of all clades of the serovar.

MALDI-TOF MS [44,45] is already applied to bacterial subtyping, but it hardly recognizes the serovar level due to the limited range of 2−20 kDa mass signals produced by the laser-dissociated proteins, such as ribosomal proteins. However, by the shotgun proteomics in our research, the identification almost covered all the annotated proteins, especially a molecular weight (MW) ≥ 15 kDa, thus it achieved an accurate serotyping. In MALDI-TOF MS, the stable mass signals are available to detect and simply pattern-match to the reference database for strain identification and similarity analysis, and no information will be provided other than molecular mass. For LC-MS/MS, the mass signals are calculated and reconstructed to peptides or proteins. By monitoring the peptides or protein markers, serotyping will be completed and a difference within one serotype could be obtained. Furthermore, with these variable peptides or proteins, especially the changed ones, we could understand the molecular mechanisms of the virulence differences and drug susceptibility of different *Salmonella* serotypes, which is helpful to explore the evolutionary rules of *Salmonella* serotypes and completely describe their adaptation to animal sources. Genetic and metabolic signatures of *Salmonella* also can be attached and improved by the proteomics network [33].

The SPSS data mining methods in this work totally extracted four peptide markers for the serotyping of *Salmonella enterica*. We hope a multiple reaction monitoring (MRM) method can be developed to quantify these target peptides for serotyping, even breakout indicating. The C5.0 predicted model in this research was proven to be effective in *Salmonella enterica* serotyping, but it needs to be further optimized in the case of an increased sample size.

The biological analysis indicates that for extra- or intra-serotypes, the changed proteins usually belong to enzymes. Therefore, we suppose that if there is a chance to develop some kind of strip holds, the test chambers containing dehydrated media, having chemically-defined compositions for each test to detect enzymatic activity, are mostly related to our reported characters mentioned above, such as the fermentation of carbohydrates or the catabolism of proteins or amino acids by the inoculated organisms. By adding the bacterial suspension to rehydrate each of the wells and incubate the strips, the metabolism produces a detectable color to, finally, monitor.

The multiple sample ANOVA test is a widely used statistic tool which has a high efficiency and is easy to spread. In this article, it was capable of distinguishing the five serotypes. For more *Salmonella enteric* isolates collected, the senior statistic methods including the Decision Tree, R-Forest, and Support Vector Machine (SVM) should be applied. Moreover, the peptides fraction strategy seems to be a better option due to its high capability. The peptide markers for different serotypes will be optimized in the future by our developed proteomic approach.

## 4. Materials and Methods

### 4.1. Bacterial Strains

A total of 40 *Salmonella enterica* strains belonging to 5 different serovars (Table 1) were collected from Chengdu Center for Disease Control and Prevention. All strains were biochemically differentiated on the subspecies level and serotyped by slide agglutination with O antigen-specific and H antigen-specific sera (Sifin Diagnostics, Berlin, DEU) according to the White–Kauffmann–Le Minor scheme [1]. The selected strains were isolated from infected humans and contaminated food. All the *Salmonella enterica* strains were grown for 24 h at 37 °C on LB agar plates (Teknova, Hollister, CA, USA).

### 4.2. Cell Lysis and Protein Extraction

Bacterial cells were centrifuged at 5000× *g* for 10 min to collect pellets, in which 5 mL of B-PER Complete Reagent (B-PER™ Complete Bacterial Protein Extraction Reagent, Thermo Scientific, Waltham, MA, USA) per gram of cell pellet was added to mix up and down. The suspension was incubated 15 min at room temperature with gentle rocking, following soluble proteins which were separated from the insoluble parts by centrifuge at 16,000× *g* for 20 min. Finally, cell supernatant was transferred to a new tube for protein concentration determination by BCA assay (Beyotime, Shanghai, China).

### 4.3. Trypsin Digestion and Peptide Enrichment

We applied the filter-aided sample preparation (FASP) [46] and stop-and-go-extraction tips (StageTips) protocols [47] for protein digestion and desalting. Briefly samples were heated for 30 min at 50 °C for reduction. YM-30 membrane filters (Merck Millipore, Burlington, MA, USA) were activated with 200 μL 100 mM NaOH, then equilibrated with 200 μL 8 M urea buffer, and centrifuged for 15 min. Samples of 200 µg were added into the filters, centrifuged, and washed two times with urea buffer. The alkylation was performed by 10 μL of 500 mM IAA in 90 μL urea buffer for 30 min at 37 °C in the dark and centrifuged. Next, 200 μL of 50 mM Ammonium Bicarbonate Buffer (ABC) was added to the filter and centrifuged three times. Subsequently, 4 μL of 0.5 µg/μL MS-grade trypsin (V5280, Trypsin Gold, Promega Corporation, Madison, WI, USA) in 100 μL ABC buffer was added to incubate at 37 °C overnight. The enzymatic digestion was stopped by centrifuging and the filter was washed by 200 μL ABC buffer again. The filtrate was selected and then subjected to SpeedVac Vacuum Concentrators (Thermo Scientific, Waltham, MA, USA) to dry out. All centrifugation steps were performed at 16,000× *g*. The sample was resuspended with 100 μL 0.2% acetic acid. By activating the self-made C_18_ embedded tips with 200 μL methanol and water, the sample was added into the tip and centrifuged at 4600× *g*. After washing the tip with 200 μL 0.2% acetic acid, the samples were eluted by using 200 μL 40% acetonitrile (ACN) in water and 400 μL 80% ACN in water. The eluent was dried out and resolved in 40 μL 0.1% formic acid, and 4 μL for LC-MS/MS analysis.

### 4.4. Nanoflow High-Performance Liquid Chromatography (HPLC) 

Peptide samples were separated by HPLC on a PicoFrit analytical column (75 μm × 10 cm, 5 μm BetaBasic C_18_, 150 Å, New Objective, Littleton, MA, USA) at a flow rate of 300 nL/min. A 130 min LC gradient was applied. The gradient started with 98% solvent A (0.1% formic acid in water), and increased to 35% solvent B (0.1% formic acid in ACN) over 110 min, followed by a steeper gradient to 80% solvent B over 15 min. 

### 4.5. MS Identification

The peptides were identified by LC-MS/MS analysis on an Ultimate 3000-nano LC apparatus and a Q Exactive mass spectrometer system coupled via a FLEX nano-electrospray ion source (Thermo Scientific, Waltham, MA, USA). Eluting peptides were sprayed at a voltage of 2.3 kV and acquired in an MS data-dependent mode using XCalibur software (version 2.2, Thermo Scientific, Waltham, MA, USA). Survey scans were acquired at a resolution of 70,000 over a mass range of *m*/*z* 300 to 1800 with an automatic gain control (AGC) target of 10^6^. For each cycle, the top 20 most intense ions were subjected to fragmentation by high-energy collisional dissociation with normalized collision energy of 27 eV. The induced fragment ions from the MS/MS scans were acquired at a resolution of 17,500 with an AGC target of 5 × 10^4^. Dynamic exclusion was set to 20 s. Unassigned ions were rejected and only those with the charge ≥2 were subjected to HCD fragmentation. 

### 4.6. Pulsed-Field Gel Electrophoresis (PFGE)

All the *Salmonella* isolates were subjected to PFGE, according to the standardized protocol of the CDC PulseNet (PNL05, April 2013). Briefly, cell suspension buffer (100 mM Tris, 100 mM EDTA, and pH 8.0) with a turbidity reading of 1 to 1.3 was mixed in equal volume with molten 2% low-melting point agarose, pipetted into disposable molds and then stored at 4 °C for 20 to 30 min. They were then incubated overnight at 56 °C in 1 mL of lysis buffer (0.5 M EDTA, 0.5 M Tris, 1% N-laurylsarcosine) with 250 μg/mL proteinase K (Promega, Madison, WI, USA). The sterile ultrapure water and 0.01 M Tris-EDTA buffer, pH 8.0 were used to remove excess reagents and cell debris from the lysed plugs. Chromosomal DNA was digested with 30 U of XbaI (Fermentas, Lithuania) for 3 h at 37 °C. Electrophoresis was carried out with 0.5× TBE buffer at 6 V/cm and 14 °C by CHEF DRIII system (Bio-Rad, Hercules, CA, USA). The running time was 20 h and the pulse ramp time was 5 to 30 s. *Salmonella enterica* serotype Braenderup, strain H9812 was used as a size marker. The gels were visualized on a UV transilluminator, and photographed by a digital imaging system (Gel Doc XR, Bio-Rad) which subsequently converted the gel images to the TIFF file format. DNA fragment patterns were analyzed with BioNumerics software (Applied Maths, Ghent, BEL). All the isolates were clustered into different pulsotypes by genetic similarity cut-off ≥85%. Reproducibility power was confirmed by comparing the fingerprint patterns that were obtained from duplicate runs of the same isolates.

### 4.7. Quantitative Proteomic Analysis and Bioinformatics Methods

Raw MS raw files were imported into the MaxQuant software suite (v1.6.0, Jürgen Cox, Munich, DEU) [48] with the default settings for quantification via MS1 peak integration and normalization of proteomic data comparing multiple samples. We used the label-free quantification (LFQ) function to estimate protein abundances in all of the analyzed samples. The parameters for database searching were set as following, including UniProt KB database (*Salmonella*, 30 August 2021), trypsin digestion with two missed cleavages, carbamidomethyl (C) as a fixed modification, oxidation (M), and acetyl (protein N-term) as the variable modification. Initial peptide mass tolerance was set to 7 ppm and fragment mass tolerance was 0.5 Da, with +2 as the default charge state of each peptide. The false discovery rates (FDRs) of peptide were both set to 0.01. An automated R-based QC pipeline called Proteomics Quality Control (PTXQC) [49] for LC−MS/MS data generated by the MaxQuant software pipeline was applied to detect measurement bias, verifying consistency, and avoiding propagation of error. PTXQC created a QC report containing a comprehensive and powerful set of QC metrics, augmented with automated scoring functions. The replicates for each sample, which failed in Alignment performance for Retention time (RT) correction (RT difference (ΔRT) to Ref > 0.7 min), would be removed for downstream analysis. The acceptable peptide results were imported into Perseus (Jürgen Cox, Munich, DEU) to perform some data transference and statistical analysis. 

### 4.8. The Predicted Model Development

The SPSS Modeler of IMB has been implemented with various tools and algorithms to model and assess the impact of peptide biomarkers on *Salmonella* serotyping. To prepare and test the models, 40 isolates were randomly categorized into two groups for model training (25 isolates) and testing (15 isolates). All 53 candidate peptides were employed as variables, the serotype group being the target.

## 5. Conclusions

Food-borne diseases caused by *Salmonella enteric serovars* represent a serious public health problem worldwide. A quick identification for the pathogens is critical for controlling food pollution and disease spreading. So far, we have developed a proteomic method to identify epidemiologically important *Salmonella enterica subsp. enterica serovars* and determined the genotypic similarity among *Salmonella enteric* isolates. Compared to the classical White–Kauffmann–Le Minor scheme and PFGE, the LC-MS/MS-based proteomic approach is of equal power, but with more specificity. Taking the 53 most variable serovar-related peptides by label-free quantitative proteomics, a C5.0 predicted model with four predictor peptides was generated with the accuracy of 94.12%.

## Figures and Tables

**Figure 1 molecules-27-04334-f001:**
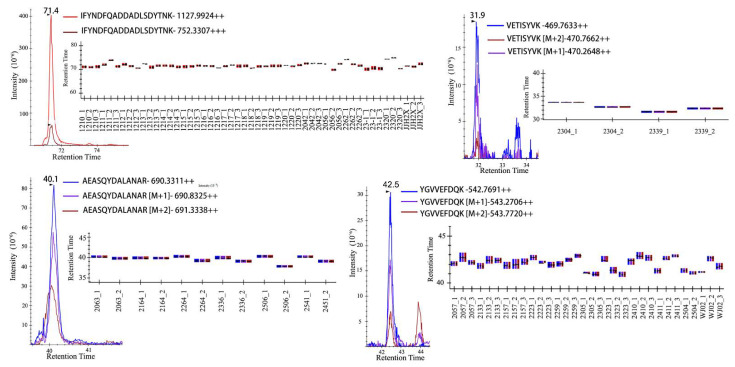
The parts of LC-MS/MS spectra of *Salmonella enterica* serovar-identifying peptide markers in training group.

**Figure 2 molecules-27-04334-f002:**
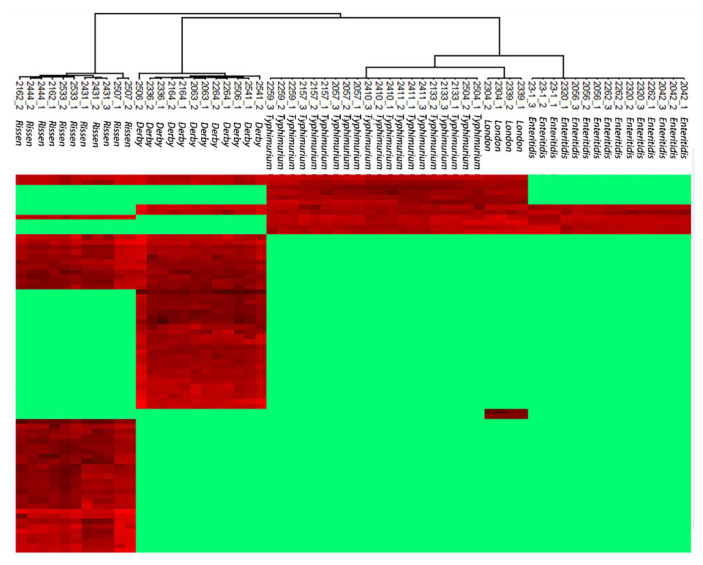
Hierarchical cluster analysis with 53 peptide markers in training group. Twenty-five isolates from five serotypes were divided into five clusters without overlap.

**Figure 3 molecules-27-04334-f003:**
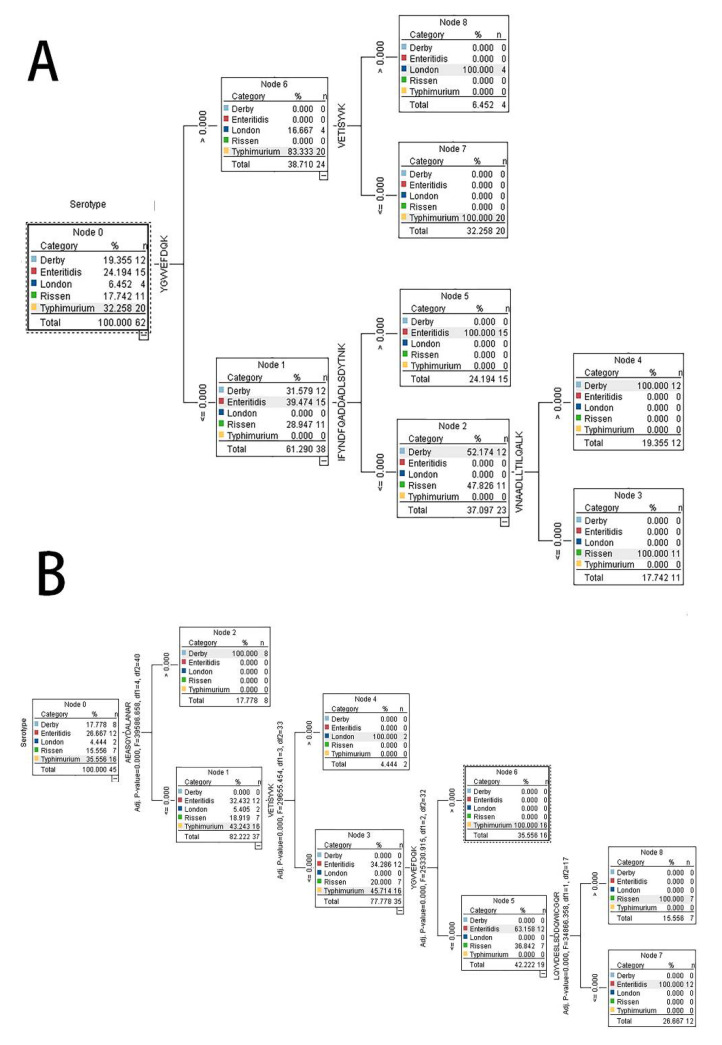
The decision tree for the prediction of *Salmonella enteric* serotypes. The model based on the C5.0 (**A**) and QUEST (**B**) method.

**Figure 4 molecules-27-04334-f004:**
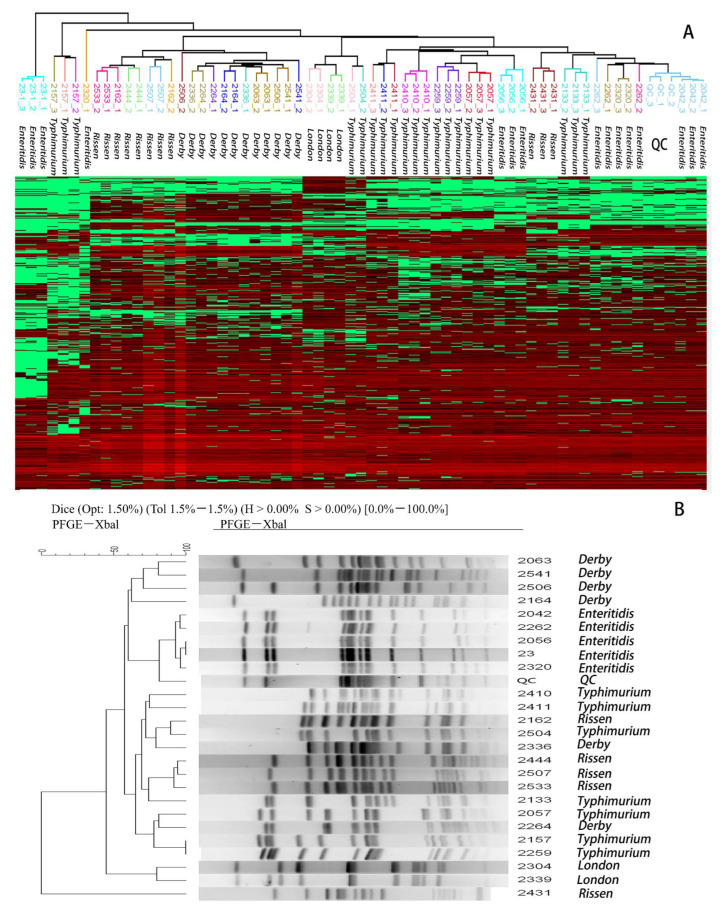
Hierarchical clustering to differentiate similarity among *Salmonella enteric* isolates. The cluster analysis by LC-MS/MS (**A**) and PFGE (**B**) in the training group. There was no evident similarity between *Salmonella enteric* strains.

**Figure 5 molecules-27-04334-f005:**
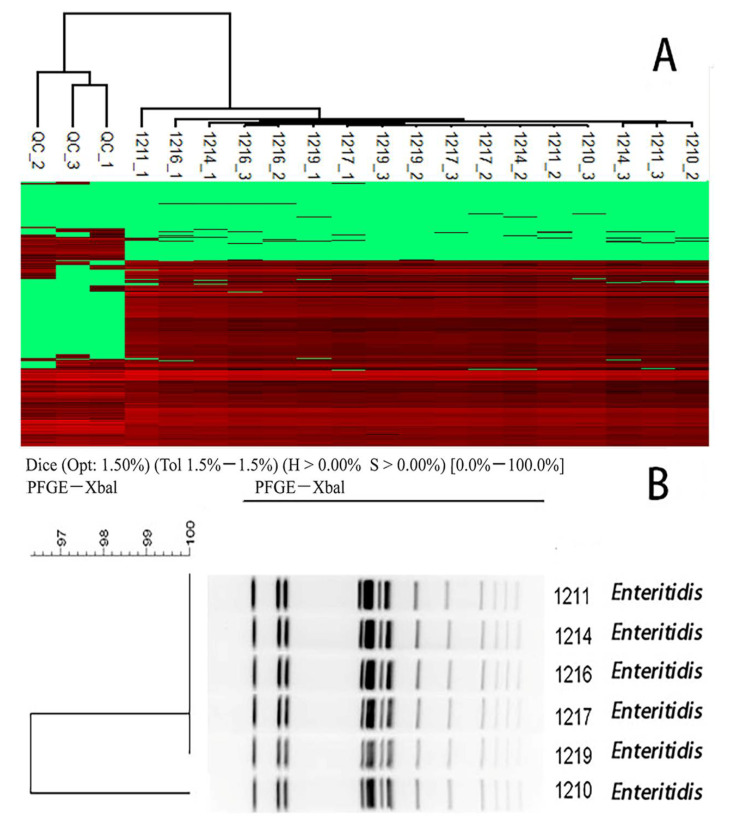
Hierarchical clustering to identify similarity among *Enteritidis* isolates. The cluster analysis by LC-MS/MS (**A**) and PFGE (**B**) in testing group. The strains No. 1211, 1214, 1216, 1217, and 1219 were certificated as the gene-closed strains in both ways.

**Table 1 molecules-27-04334-t001:** *Salmonella enterica subsp. enterica* strains (*n* = 40) used in this study.

Group	Serovar	No. of Strains	Source(s)
	*Enteritidis* (9,12:g,m:-)	5	Human, food
	*Typhimurium* (4,5,12:i:1,2)	7	Human, food
Training	*Derby* (4,5,12:f,g:-)	6	Human, food
	*Rissen*(6,7:f,g:-)	5	Human, food
	*London* (3,10:l,v:1,6)	2	Human, food
	*Enteritidis* (9,12:g,m:-)	6	Human, food
	*Typhimurium* (4,5,12:i:1,2)	3	Human, food
Testing	*Derby* (4,5,12:f,g:-)	2	Human, food
	*Rissen* (6,7:f,g:-)	2	Human, food
	*London* (3,10:l,v:1,6)	1	Human, food
	*Sagona*. (4,5,12:f,g,s:-)	1	Human, food

## Data Availability

The data presented in this study are available on request from the corresponding author.

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
