# Peer review of "Proteome-Based Serotyping of the Food-Borne Pathogens Salmonella Enterica by Label-Free Mass Spectrometry"

_molecules, 2022, doi:10.3390/molecules27144334_

Round 1

Reviewer 1 Report

The paper provides a simple and efficient method for a pathogen serotyping. The general idea and results provide a reliable platform for high-throughput strain profiling.

Several issues should be resolved:

1) The quality of figures should be enhanced.

2) CDDs and proposed functions of proteins should be added to the SI table or somewhere.

3) The proposed functions of enzymes and their natural variability should be discussed more extensively. 

Reviewer 2 Report

Dear authors, I consider that your work is relevant and represents a novel application and use of protein mass spectrometry in the area of clinical diagnosis, however, there are some important aspects that must be addressed. 

It is important to point out that the work does not represent a peptidomic study, since in peptidomics the endogenous forms of the peptides in a sample are identified by mass spectrometry, that is, the peptides that are already in the biological system from the beginning. In this case, since a protein extraction was carried out, which was then hydrolysed to generate peptides and analysed by mass spectrometry, it is actually a shotgun-type proteomic approach.  

The discussion does not offer a contrast of results with previous studies or with techniques already implemented for diagnosis in terms of specificity, sensitivity or ability to discriminate between serotypes with respect to other approaches. 

Images are of meagre resolution and quality. It is recommended to redesign or resize them in such a way that the details can be appreciated. 

Line 16: Please change “Salmonella enteric” by “Salmonella enterica”

 Please homogenize the format of units, for example: in Line 266 “16,000 × g” is used, while in Line 281 appears as “16000g”. Also, verify the spaces between numbers and units through the text.

Reviewer 3 Report

I have read the manuscript and decided to reject it for publication. Below I explain the reasons.

The authors propose a technique (LC/MS-MS) that broadly leads to the same results that were obtained from a PFGE analysis, arguing that the first one is faster, but they do not mention how much faster, nor do they talk about the cost-benefit. Given my knowledge, the first technique will not save a significant amount of time, and it is not entirely cheap either. In line 359-360, the authors talk about the speed, but the question would be why look for a technique that will give me the same information as another that already exists that is also fast and a little cheaper. If LC/MS-MS analysis will provide more information, this could be a reason to consider it.

On the other hand, there are techniques such as MALDI-TOF and MALDI Bio Typer, which have already been described and published for the same purposes, which are also fast and accurate, here a couple of references (10.1371/journal.pone.0040004, 10.1128/JCM.02590-15).

Finally, the figures are of poor resolution, difficult to analyze and there are multiple formatting errors.

Round 2

Reviewer 2 Report

No comments

Reviewer 3 Report

Thanks for trying to answer and follow the suggestions.